# Blood Pressure and Heart Rate Responses to an Isokinetic Testing Protocol in Professional Soccer Players

**DOI:** 10.3390/jcm11061539

**Published:** 2022-03-11

**Authors:** Arturo Pérez-Gosalvez, Francisco García-Muro San José, Ofelia Carrión-Otero, Tomás Pérez-Fernández, Luis Fernández-Rosa

**Affiliations:** Facultad de Medicina, Universidad San Pablo-CEU, CEU Universities, Urbanización Montepríncipe, Boadilla del Monte, Boadilla, 28660 Madrid, Spain; apgosalvez@ceu.es (A.P.-G.); o.carrion@ceu.es (O.C.-O.); tpfernan@ceu.es (T.P.-F.); luferro@ceu.es (L.F.-R.)

**Keywords:** soccer, blood pressure, heart rate, isokinetic dynamometry, testing, treadmill test

## Abstract

The aim of this study was to determine blood pressure (BP) and heart rate (HR) responses triggered during an isokinetic testing protocol in professional soccer players and compare cardiovascular parameters at completion of this isokinetic protocol with those during a treadmill test. Using purposive sampling, 63 professional soccer players were recruited. Cardiovascular responses were measured noninvasively during a bilateral testing protocol of knee flexion and extension. Treadmill ergospirometry following an incremental speed protocol was performed to analyze the same cardiovascular parameters at rest and at completion of this test. There were significant differences in diastolic blood pressure (DBP) and HR according to field position. The parameters presented high homogeneity at both competitive levels. Systolic blood pressure, mean arterial pressure, HR, and rate pressure product at completion of the treadmill test were significantly higher than those at completion of the isokinetic protocol. Intermittent isokinetic testing protocol of the knee triggers normal and safe BP and HR responses in healthy professional soccer players. The HR of the defenders was higher than those of the forwards and midfielders but was independent of the competitive level. The values of cardiovascular parameters at isokinetic protocol completion were lower than those during the treadmill test.

## 1. Introduction

Soccer is the most practiced sport worldwide today, with approximately 265 million players across five continents, which is equivalent to 4% of the world’s population [1]. Traditionally considered [2,3] a discontinuous and intermittent physical exercise that encompasses low-, medium-, and high-intensity activities, soccer generates physiological and metabolic demands [4] that have increased with the physical demands necessitated by the professionalization of this sport.

Researchers have studied the physiological variables of professional soccer players during matches, such as distance covered (10–13 km), [5] mean heart rate (HR; 165–170 bpm, equivalent to 85% of the maximum HR), or oxygen uptake (45–65% of VO_2_ max with telemetric controls) [6,7]. These variables are used to analyze energy metabolism, with some researchers reporting 70–85% aerobic metabolism use and the remainder as anaerobic metabolism [8]. However, this measurement is difficult because it depends on the duration of the high-intensity phases and the recovery period [9,10].

These physiological parameters have become more relevant as the requirements of professional soccer players have increased, and several cases of soccer players experiencing cardiovascular events during matches have been reported in recent years [11], with “sudden death” being the most remarkable both clinically and socially. In fact, between 25% and 49% of athletes who experienced sudden death in Spain were practicing soccer [12,13].

Therefore, many researchers have performed various functional assessment tests in the laboratory and controlled field tests that attempt to reproduce the conditions that a soccer player is subjected to during a match [8,14,15,16]. The most widely used field tests are categorized as aerobic, anaerobic, and specific, and the most referenced laboratory tests are ergospirometry, anaerobic tests, isometric/isotonic contractions, and isokinetic dynamometry [17].

Many researchers have conducted different functional assessment tests in the laboratory and controlled field tests that try to mimic, in as standardized and objective a manner as possible, the conditions that a football player is subjected to during a match [8,14,15,18]. Among all these tests, isokinetic dynamometry allows for an objective assessment of a football player’s muscle function as well as his response to maximum intensity requirements [19,20]. Thus, isokinetic dynamometric systems have been used to perform specific strength training, rehabilitate postsurgical musculoskeletal processes [21,22], prevent muscle imbalances that are a risk factor against muscle injuries [23,24,25], and evaluate the muscle strength and power of the lower extremities in soccer players [26,27]. For this, isokinetic dynamometry has been considered the gold standard among the strength tests that can be performed on a soccer player [28].

The maximum requirement of this test is an adequate musculoskeletal state and a sufficient cardiorespiratory condition to satisfy the requirements; thus, it would be interesting to evaluate a soccer player’s cardiovascular response during a usual isokinetic protocol. This protocol constitutes a controlled laboratory test, which is part of the usual physical assessment of soccer players, and generates metabolic, muscular, and cardiovascular demands that differ from those generated during maximal aerobic exercise. This cardiovascular response has previously been studied in incremental dynamic exercise in soccer players, with the treadmill test used to evaluate cardiorespiratory fitness; this method has also been used to detect possible cardiovascular functional risks [29]. Although the isokinetic assessment test is an indispensable requirement in the evaluation of the physical fitness of soccer players, there are no previous studies assessing the cardiac and blood pressure risk in these athletes.

However, to date, no reliable published studies with an adequate sample size have described how soccer players’ BP and HR respond to the demands of an isokinetic testing protocol of the lower extremity musculature. Studies describing the cardiovascular response to isokinetic exercise in healthy or old adults are scarce [30,31] and possess high sample variability [32,33,34,35], which prevents the extrapolation of their results to a highly trained population capable of developing high levels of muscle strength and power in the lower extremities.

This study aimed to describe the BP and HR responses triggered by an isokinetic testing protocol in professional soccer players and compare cardiovascular parameters at completion of this isokinetic protocol with those during a treadmill test.

## 2. Materials and Methods

### 2.1. Participants

A minimum sample size of 46 was deemed to be representative of the adult population using GRANMO version 7.12 [36], assuming a reference population of 1000, a 95% confidence interval level, an estimate of the standard deviation of 20, a precision of 6, and a 10% replacement rate. A total of 63 professional male outfield soccer players (age 22.6 ± 1.2 years; height: 179.2 ± 5.2 cm; weight: 72.9 ± 5.5 kg), including 23 defenders (age 22.3 ± 3.5 years; height 181.5 ± 4.4 cm; weight: 74.6 ± 4.1 kg), 25 midfielders (age 20.9 ± 2.5 years; height: 178.6 ± 4.7 cm; weight: 73.4 ± 5.1 kg), and 15 forwards (age: 19.7 ± 1.9 years; height: 176.8 ± 5.9 cm; weight: 70.2 ± 6.6 kg), participated in this study. They belonged to a professional soccer team during the measurement period, with 20 on the first-division team and 43 on the second-division team.

This study was performed during preseason, and players were recruited using purposive sampling. Prior to the study, the players were evaluated by a sports medicine specialist who performed an anamnesis and clinical examination following the Union of European Football Associations recommendations [37] to confirm that the players met the conditions for professional sports practice. Subsequently, electrocardiogram, echocardiography, and spirometry were performed to rule out possible contraindications in performing the tests [38]. None of the participating players showed warning signs that contraindicated their inclusion in the study.

All players who had a current federation record as a professional soccer player and who were physically and medically fit to start the season were included. Players who had undergone surgery in the previous 12 months of any pathology in the lower extremities or who had suffered an injury in the lower extremities that would have forced them to suspend sports activity for at least 1 month were excluded. Players undergoing medical treatment with a drug that interfered with physical capacity and those who suffered from acute systemic diseases were also excluded.

All participants were fully informed about the protocol prior to participation and provided informed consent in accordance with the principles of the Declaration of Helsinki. The study was approved by the ethics committee of CEU San Pablo University (approval code, 238/17/18).

A repeated-measures design involving active soccer players was used to determine the BP and HR responses triggered by the isokinetic testing protocol. Then, we compared the HR and BP values at completion of this isokinetic protocol with those achieved during a treadmill test.

Both tests (isokinetic and treadmill) were performed during the preseason of the soccer teams; the usual schedule followed in the medical and functional examinations of the soccer players’ sports club was respected. These athletes are evaluated annually using isokinetic assessment and treadmill ergospirometry of the lower extremities.

Once their medical history was recorded, the soccer players performed an isokinetic strength test in the Research Unit in the Physical Therapies Laboratory of the Faculty of Medicine of CEU San Pablo University. After completion, the data were collected, and the second test was conducted within 3 days. The treadmill test was performed in the exercise physiology laboratory of the School of Sports Medicine in Complutense University of Madrid in the recommended environment [39]. Each player’s information was encrypted to guarantee anonymity.

### 2.2. Isokinetic Testing Protocol Description

The isokinetic test was selected, as it is a test included in the functional assessment of elite football players worldwide and validated by UEFA [33] and Spanish Association of Football Team Doctors [40]. For this reason, this population group must face this test as a regular evaluation element of their physical condition.

Initially, with the participant sitting with their feet on the ground, baseline measurements of BP and HR were recorded using a BTL-08 ABPM II portable and digital sphygmomanometer (BTL Industries Ltd. Hertfordshire, United Kingdom). Subsequently, without removing the BP cuff from the participant’s upper limb, the device was placed inside a sheath that was attached to an adjustable strap placed around the waist of the participant; thus, the following measurements could be performed without replacing the sphygmomanometer.

The participants performed a general 10 min warm-up exercise on a Monark cycle ergometer (model 818E) at a moderate pace and resistance, immediately after which BP, HR, systolic blood pressure (SBP), diastolic blood pressure (DBP), rate pressure product (RPP) (RPP = HR × SBP), and mean arterial pressure (MAP) (MAP = DBP + (0.333 × [SBP − DBP])) were recorded.

The participants were then asked to remove their shoes and sit on the IsoMed2000 strength-testing system (D&R Ferstl GmbH, Hemau, Germany) in an upright position, with 85° flexion at the hip. The participants were secured by straps around the chest, waist, and right thigh. The dynamometer lever arm was aligned with the participant’s tibial spine, 2.5 cm proximal to the right medial malleolus; after ensuring it was comfortable for the participant, it was fixed using a Velcro strap. The rotating axis of the dynamometer was aligned with the knee joint’s axis of rotation (lateral epicondyle of the femur). After checking the strap tension, the participant was instructed to hold on to the hand grips on the side of the seat during the efforts [41], and the isokinetic testing protocol was initiated (Figure 1).

Before starting the protocol, the right lower extremity (RLE) was weighed using the automatic limb weighing system of the dynamometer to adjust for the gravitational effect on torque.

A bilateral study protocol of continuous concentric/concentric contraction was followed at low (60°/s), medium (180°/s), and high velocities (240°/s) of knee flexion and extension through the knee’s range of motion, from 0° (full extension) to 90° (flexed), as recommended by the Spanish Association of Football Teams Physicians in its protocol for professional soccer players [40].

Prior to assessment at each angular velocity, the participants performed three to five submaximal contractions of increasing intensity (25%, 50%, and 80%), completing the established range of motion with both knee flexion and extension, to adapt the musculature to the effort requested later [42,43]. The participants were oriented to avoid the Valsalva maneuver and to breathe spontaneously throughout the movement [44].

The protocol series required the performance of 5 repetitions of flexion and extension at 60°/s, 10 repetitions at 180°/s, and 25 repetitions at 240°/s, all at maximum intensity and always starting with the right leg. During the protocol, encouragements by verbal coaching and visual feedback were provided to all participants to help them concentrate on the quality and maximum intensity of their movements [45,46].

Immediately after completing each of the six established series of contractions, the investigator activated the BP monitor and asked the participant to rest and without speaking to check BP, HR, SBP, DBP, RPP, and MAP (approximately 30–40 s). After verifying that the device had recorded the measurements, muscle warm-up/adaptation contractions were started in the next series of protocol contractions. Between the third (right leg at 240°/s) and fourth (left leg at 60°/s) series, the time for measuring BP and HR was used to move the dynamometer lever to the opposite side of the participant. This procedure lasted 30–40 s to homogenize the rest time with that of the series performed in the right leg [47,48].

Once the final BP and HR measurements were recorded and checked (after the sixth series), the straps, attachment to the mobile arm, and sphygmomanometer cuff were removed, and the athlete was recommended to spend 5–10 min passively stretching the lower extremity musculature before leaving the investigation unit.

### 2.3. Treadmill Ergospirometry Test Description

Treadmill ergospirometry test was performed after the isokinetic test. The “incremental speed protocol” for ergospirometry commonly used in the School of Sports Medicine of Complutense University was followed in the functional assessment of the participants [49,50]. For this, a treadmill (H.P. Cosmos), gas analyzer (model Vmax, Sensor Medics), and electrocardiograph (Quest Exercise Stress System, Burdick Inc; Milton, WI, USA) were used. After checking the BP and HR of the participants at rest, they performed a 2 min warm-up exercise on the treadmill at 4 km/h following the usual protocol [51,52], with electrocardiographic control but no respiratory control.

The maximum treadmill test was started at an initial speed of 6 km/h for all participants. Every 2 min, the speed was increased by 2 km/h until the participant was exhausted, with the slope constant at 1% as in the warm-up exercise. BP and HR were measured within 30 s after completion of the exhaustion test. Recovery started at 5 km/h, and this speed was maintained until the participant’s complete recovery.

### 2.4. Statistical Analyses

We used the Kolmogorov–Smirnov and Levene tests to assess the normality of the values and the equality of variances, respectively. We then performed univariate repeated-measures analysis of variance (ANOVA) to determine differences among SBP, DBP, MAP, HR, and RPP values. Finally, we used Student’s paired *t*-test to assess possible differences between phases. Bonferroni’s post hoc tests were applied for comparative analyses of between-group differences when a significant interaction was found. SPSS version 24.0 for Windows was used for statistical analysis (Statistical Package for the Social Sciences, Chicago, IL, USA). The results are expressed as the mean ± standard error, and the significance level was *p* < 0.05.

## 3. Results

### Descriptive

A global descriptive analysis of the cohort (n = 63) including the different cardiovascular parameters at the different measurement times is shown in Table 1.

During the isokinetic protocol, the maximum SBP value was 207 mmHg (at the fourth measurement), maximum DBP was 103 mmHg (at the third measurement), and maximum HR was 148 bpm (at the final measurement). The minimum SBP, DBP, and HR values were 102 mmHg, 44 mmHg, and 39 bpm. Intra-subject differences in DBP with respect to the measurement at rest never exceeded 13 mmHg.

Because all the parameters described followed normal distribution in the cohort (n = 63), repeated-measures ANOVA was performed to compare each parameter among the different measurement points and to assess the effect of time on the obtained means (Figure 2).

ANOVA revealed an effect of measurement time on SBP (F_5.363_ = 52.91; *p* < 0.001; ɳp2 = 0.5). When comparing pairs of measurements, a significant effect was produced at rest and after 10 min warm-up exercise compared with that at the other measurements (*p* < 0.001). There were no significant differences between the third measurement (post RLE 60°/s) and the other measurements or between any of them (*p* > 0.05). DBP was also affected by different measurement times (F = 9.30; *p* < 0.001; ɳp2 = 0.149). The pairwise comparison showed statistically significant differences between the measurement at baseline and those at rest except for the sixth (*p* = 0.313) and final (*p* > 0.5) measurements. There were also significant differences between the third and fourth measurements (post RLE 60°/s and post RLE 180°/s) with respect to the sixth measurement (*p* = 0.045 and *p* = 0.016, respectively) and between the fourth and final measurements (*p* = 0.03). HR was also affected by different measurement times (F_3.521_ = 188.37; *p* < 0.001; ɳp2 = 0.780). HR resulted in highly significant differences (*p* < 0.001) among all the measurements except between the fifth (post RLE 240°/s) and seventh (post LLE 180°/s) measurements.

Finally, the effect of the different measurement times on MAP (F_5.538_ = 29.47; *p* < 0.001; ɳp2 = 0.357) and RPP (F_4.577_ = 168.55; *p* < 0.001; ɳp2 = 0.761) were observed. There were significant differences in MAP between the baseline measurement with respect to the rest measurement (*p* < 0.001) and between the second and fourth measurements (*p* < 0.01). There were differences in RPP among all measurements except among the fifth, sixth, and seventh measurements (*p* > 0.05) and between the fourth and sixth measurements (*p* = 0.086).

Repeated-measures ANOVA was performed to assess the effect of time on cardiovascular parameters in the participants according to their field position (defenders, midfielders, and forwards) as well as the interaction of this effect with field position.

The changes in SBP (F_10.745_ = 1.02; *p* = 0.463) and RPP (F_9.252_ = 1.66; *p* = 0.096) with respect to the measurement points followed the same profile as that in the global sample. There was no interaction of the effect “time” and the factor “field position” when comparing the means of the different measurement points. An interaction of the effect of time with respect to field position was obtained for DBP (F = 2.1; *p* = 0.012; ɳp2 = 0.076), with a significant difference at the fourth measurement (180°/s RLE) between the defenders and forwards (*p* = 0.043). Although there were no significant differences among the field positions in the other measurements, the DBP response at the fifth measurement differed among the groups. In turn, an interaction effect of “time–field position” was found for HR (F_7.086_ = 1.76; *p* = 0.042; ɳp2 = 0.065) during the isokinetic protocol measurements. The mean HR values of the defenders were higher than those of the forwards and midfielders, with significant differences at the sixth measurement (*p* = 0.037); the other measurements displayed homogeneous HR response in all groups. Finally, there was an interaction effect of “time–field position” in MAP (F_14_ = 1.96; *p* = 0.02; ɳp2 = 0.072), although when performing pairwise comparison, no significant differences were found among the groups at any of the measurements. MAP in the forwards changed with respect to that in the defenders and midfielders, without significant differences.

When the participants were categorized according to competitive level (first- and second-division teams), the cardiovascular parameters during the isokinetic protocol followed a normal distribution. No interaction of “time effect” and “category factor” in the cardiovascular parameters was recorded in the comparison of the means at the different measurement points. SBP (F_5.361_ = 0.31; *p* = 0.914), DBP (F = 0.48; *p* = 0.84), HR (F_3.559_ = 0.78; *p* = 0.520), MAP (F_5.507_ = 0.34; *p* = 0.9), and RPP (F_4.601_ = 0.67; *p* = 0.62) showed high homogeneity in their response in both categories.

Table 2 shows the mean values of the cardiovascular parameters analyzed at the end of both tests and the significance of the comparisons.

There were significant differences in the final SBP, MAP, HR, and RPP values of both assessment tests, with all parameters significantly higher at the end of the treadmill ergospirometry.

## 4. Discussion

Notably, none of the participants presented complications or abnormal BP or HR responses to the exercises [53]. The cardiovascular reference values used to determine abnormal responses were those used for dynamic incremental exercise (treadmill test) [38,54,55] because no study in the literature has established non-physiological BP and HR responses for isokinetic exercises. Therefore, because “normal” cardiovascular response values during an isokinetic testing protocol have not been established, it is advisable to use those established for stress tests as a fundamental reference when studying cardiovascular responses during maximum exercise.

The novelty of this study lies in describing the behavior of BP and HR in the group of professional soccer players not because they are just another population group but because they will be subjected to this assessment test on a regular basis during the seasons in which they compete. This means that there is a high prevalence of this test in this specific population group. Furthermore, this test could be used in the future to obtain more global and detailed information on the AT and HR of a soccer player to these physical demands and prevent possible future undesirable or pathological clinical events.

There were no players in whom SBP decreased in relation to that at rest, which is recommended [56]. This is considered a normotensive response to the effort because maximum SBP and DBP values of up to 240 and 115 mmHg, respectively, have been established in highly trained participants [57,58], with an increase in DBP of up to 15 mmHg considered normal during maximum-intensity exercise [54,59]. The maximum HR recorded was clearly below the cardiovascular safety limits as expected from intense exercise with limited duration.

### 4.1. Heart Rate

The mean HR of all participants showed a practically linear significant increase throughout the assessment protocol except between the fifth and sixth measurements, with differences between each consecutive recording time of approximately 7–13 bpm, and the mean HRmax was 112.9 ± 18.9 bpm at completion of the protocol. It is evident that the interruption in the linear increase in HR at the sixth measurement (LLE 60°/s) is related to the change in the LE made by muscular effort, subsequently continuing the progressive increase in HR with the same profile as before the change of LE. In fact, it is probable that if the six series of isokinetic testing were performed with the same LE, higher maximum values than those obtained in this study would be achieved.

On analyzing the variations in HR between each proposed angular velocity, we observed that the greatest increases occurred after warming up on an exercise bike (13.4 bpm) and in the series at high angular velocity (240°/s), where it increased by 13.1 bpm (RLE 240°/s) and 11.6 bpm (LLE 240°/s) compared with that in the immediately previous series measurement. In these series at 240°/s, 25 repetitions of knee flexion/extension were performed, leading to a longer effort time than that in the series at 60°/s (5 repetitions) and 180°/s (10 repetitions). This finding corroborates the results of other studies, in which the increase in HR depends more on the duration than on the intensity of the exercise [31,60,61,62]. Because isokinetic testing protocol is designed with consecutive series and little recovery time [30,60], it results in higher HR responses than those to isolated series at a given angular velocity or at rest intervals greater than 90 s between each series [35,63]. In fact, in a continuous isokinetic testing protocol, the mean HR increased to values close to those obtained in maximum stress tests until the participant reaches exhaustion [64].

However, in the usual isokinetic tests of the knee musculature in soccer players [27,65,66], no more than three or four series are performed at different velocities, so the effort of each series does not exceed 1 min; therefore, HRmax values are not attained. Studies that report greater increases in HR in adults after a series of isokinetic exercises generally have longer durations of the said series [30,67].

On comparison of the HR response of our sample during the isokinetic testing protocol with that in the studies, it is evident that HR in soccer players is much lower than that in untrained adults [30,31,34,62,68]. This suggests that cardiovascular adaptations to soccer players’ training trigger a smaller increase in HR during isokinetic exercises despite our sample having a lower mean age than that of samples in other studies. Given that higher HR values are obtained during isokinetic exercises in young individuals [30,35,68], the adaptations to training by this population group are more significant than the participants’ age.

This reflection seems to be confirmed by the absence of differences in the HR response during isokinetic exercises between the first-division team (24.5 years) and second-division team (19.9 years) players. In fact, the behavior of HR between both groups was very similar at all measurements, indicating the limited influence of age and competitive level in the HR response.

### 4.2. Blood Pressure

The SBP values of the global sample increased progressively until the fourth measurement (RLE 180°/s), when, after reaching a mean value of 155.2 ± 15.7 mmHg, it remained practically unchanged at the subsequent measurements. It even decreased slightly at the seventh and eighth measurements until completion of the isokinetic testing protocol, with a mean value of 154.3 ± 15.3 mmHg. After the third and fourth measurements, there were no significant differences in the increase in SBP, resulting in an incremental curve that reached a plateau (fourth measurement), and it remained stable until completion of the protocol. This SBP response is like that described in healthy adults performing compared to incremental dynamic exercises [69,70] although the mean values in our study were lower. Similarly, no decrease in SBP was observed after changing LE between the fifth and sixth measurements as in HR, so this change did not influence the overall response to the protocol.

We did not identify an influence of angular velocity on the BP response because the increases occurred during the first two series, and SBP subsequently remained unchanged. Some researchers who assessed the SBP response in isokinetic exercise series at different velocities recorded higher SBP at low angular velocities [32,34,67,71], whereas others did not report significant differences between knee flexion–extension series at different angular velocities, as in our case [60,62]. Therefore, it appears logical that in consecutive series of exercise protocols with limited recovery time, angular velocity is not a relevant element in the SBP response.

The SBP values in this study are hardly comparable with those in other studies because no similar designs were found that measured cardiovascular parameters during an isokinetic testing protocol, and no studies assessed professional soccer players [34,72]. Thus, the lower SBP values in our participants are likely related to better cardiovascular adaptations to exercise by soccer players; the BP response profile may follow a similar pattern in healthy adults. It is evident that an isokinetic protocol with a series of contractions established at different velocities cannot generate excessive increases in SBP as reported during protocols performed to exhaustion [32] or with heavy-resistance exercises involving large muscle groups [73,74].

The position of the soccer players was not associated with differences in the SBP response although the midfielders had higher baseline values (130.8 mmHg) and maintained them during essentially all measurements compared with the other players, with a non-significant increase in final SBP of 5 and 7 mmHg compared with that in forwards and defenders, respectively. However, these small differences do not follow a stable pattern that justifies an influence of the field position on the SBP response. These differences are even smaller when comparing players based on their competitive level, in which both the SBP response and the mean values obtained by the two groups are very similar and not significantly different.

The DBP value increased by 8 mmHg after warming up and remained almost unchanged until the fourth measurement (RLE 180°/s), after which it gradually decreased except for a slight increase in the seventh measurement, reaching a mean value of 74.9 ± 9.9 mmHg at protocol completion. This slight increase in DBP is lower than that reported in other isokinetic (non-exhausting) exercise designs in untrained participants [75,76] and clearly lower than that in studies with isometric exercise protocols for HR or percentage of VO_2_ max [77,78,79]. Thus, the adaptation to exercise by individuals with a high level of training seems to trigger lower values of DBP response during isokinetic exercises. Notably, the expected DBP response to non-exhausting isokinetic exercise protocols is a slight increase of ≤15 mmHg in highly trained participants. This behavior differs from that to dynamic incremental exercises [50] but is very similar to that to non-incremental exercises [80]; therefore, the duration and particularly the progressive intensity of the exercise seem to be key elements in the behavior of this parameter.

The greatest increase in DBP occurs after warming up, and it remains largely unchanged thereafter. Therefore, it is evident that the possible hypertensive effect related to isokinetic exercise would only be associated with exhausting isokinetic exercise designs [32,64] as in the case of isometric exercise [79,81].

The DBP response patter was very similar in both groups of soccer players, with no differences greater than 3 mmHg in the mean values. No influence of age or competitive level was noted in the observed response although some studies involving untrained healthy individuals reported a slight dependence of age on this response during isokinetic exercise [30,68]. This effect is decreased in highly trained individuals; thus, the DBP response is determined by the adaptations to training by soccer players.

However, certain variations in the DBP response according to field position were observed. In general, defenders had DBP throughout the protocol, with the forwards reporting lower values until the sixth measurement, after which they exceeded the mean DBP of the midfielders. These results differ from those of SBP; i.e., midfielders had higher SBP than forwards and defenders; at protocol completion, the differential BP of the midfielders (86 mmHg) was higher than that of the forwards (79.3 mmHg) and defenders (74 mmHg). These findings agree with those of previous studies that evaluated other types of exercises [82,83]; there is a linear relationship in which as the subject’s training level, maximum TAS, and differential BP increase. Thus, due to the physical demands of their position, midfielders may have better BP adaptation to an intermittent protocol of isokinetic exercises at various velocities. However, this aspect is not recorded in continuous incremental aerobic exercises, as reflected in our ergospirometry results or those of Ramos [50], resulting in greater differences according to field position when faced with high-intensity intermittent efforts (isokinetic testing) than when performing a continuous incremental aerobic effort. This may all be influenced by an increasing interest in improving the aerobic capacity of field soccer players [84,85,86,87] regardless of their position, whereas adaptation to aerobic–anaerobic efforts are determined to a greater extent due to the demands of the footballer’s position during competition.

Finally, MAP and RPP were determined because TAM has been previously used to assess BP response during isokinetic exercise [31,68,76,88], and RPP allows us to clinically objectify myocardial O_2_ consumption during the test [89,90].

MAP progressively increased until the fourth measurement, with the cycling warm-up clearly resulting in a more marked increase in TAM (10.4 mmHg) and with values clearly lower than those recorded in other designs of isokinetic exercise both in young adults [68,76,88] and older, untrained subjects [31,68] No differences were observed in MAP according to the mean age of the participants in a previous study [68] similar to our results. There was no influence on the type of contraction selected because in general, a series of concentric contractions are considered more “hypertensive” than eccentric contractions at the same angular velocities [31,68,91] However, this influence on the MAP response according to the type of contraction selected appeared to be related only to samples from untrained and mainly older subjects, in which concentric-type isokinetic exercises trigger higher MAP, SBP, and DBP [31,68].

Regarding field positions, a slightly different MAP response was observed in the forwards, who had increased MAP in the final two measurements compared with those in the previous measurements, which, although not significant, showed increased BP near the end of the effort. This may be related to the type of physiological effort they usually perform, such as short and intense efforts but with longer recovery times; thus, an intermittent isokinetic testing protocol would reflect more differences in BP response according to the field position than commonly used treadmill tests.

However, this phenomenon was not reported for RPP, in which the behavior of the players was very similar in all groups. The clinical estimate of myocardial O_2_ consumption that results from this parameter [89,92] would show a very homogeneous behavior among the groups of soccer players. The RPP values in our study are similar to those of other researchers who assessed this parameter in untrained participants [31,68,93] and with designs of isolated isokinetic exercises or with rest intervals greater than 90 s between each series, which results in higher BP and lower HR. Findings similar to the maximum RPP values in our participants during the isokinetic protocol were 15.000–17.000 units lower than those obtained by professional soccer players in an ergospirometry in previous studies [14,50].

This study shows at least one limitation. The between-groups and between-categories results are not strong. Some commonly utilized physiological parameters (e.g., HR or VO_2_ max.) may not be sensitive enough to detect specific physiological adaptations occurring in response to fatigue/training [94,95,96]. One possible explanation for this may stem from the fact that these parameters provide little information on the specific nonlinear dynamic interactions between organic subsystems involved in exercise physiology [95]. Therefore, it would be interesting to evaluate the effects of isokinetic/treadmill protocols utilizing variables able to quantify how respiratory, cardiovascular systems, and neuromuscular systems coordinate during exercise in future studies.

## 5. Conclusions

The findings indicate that the performance of an intermittent isokinetic testing protocol of the knee triggers normal and safe BP and HR responses in healthy professional soccer players, with no values exceeding the recommended cardiovascular stability limits.

The angular velocity is not a determining element in the SBP and DBP response.

HR increased linearly during the isokinetic testing protocol until reaching submaximal values, and its increase depends to a great extent on the duration of the isokinetic effort than on its intensity.

The HR of the defenders was higher than those of the forwards and midfielders but was independent of the competitive level.

The SBP and HR values achieved at completion of the treadmill test were significantly higher than those during the isokinetic testing protocol. The final DBP in the isokinetic protocol was higher than that measured at completion of the treadmill test, but the results were not significantly different.

## Figures and Tables

**Figure 1 jcm-11-01539-f001:**
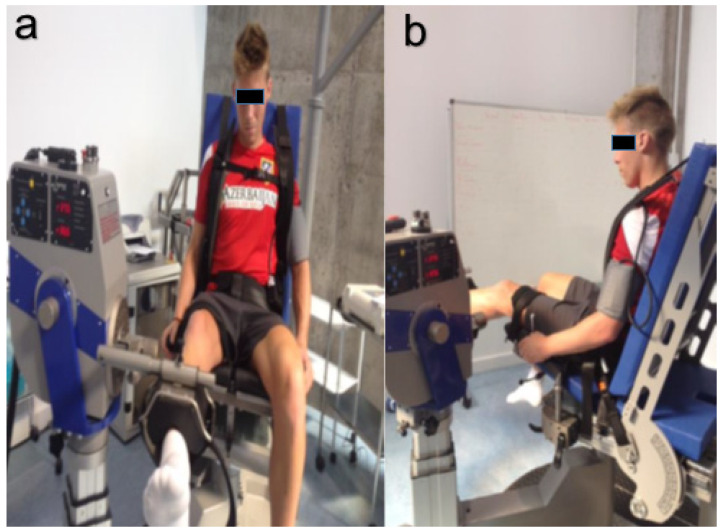
Positioning of a subject before starting the isokinetic testing protocol (**a**) and prior to the beginning of the contraction series with the left lower extremity (**b**).

**Figure 2 jcm-11-01539-f002:**
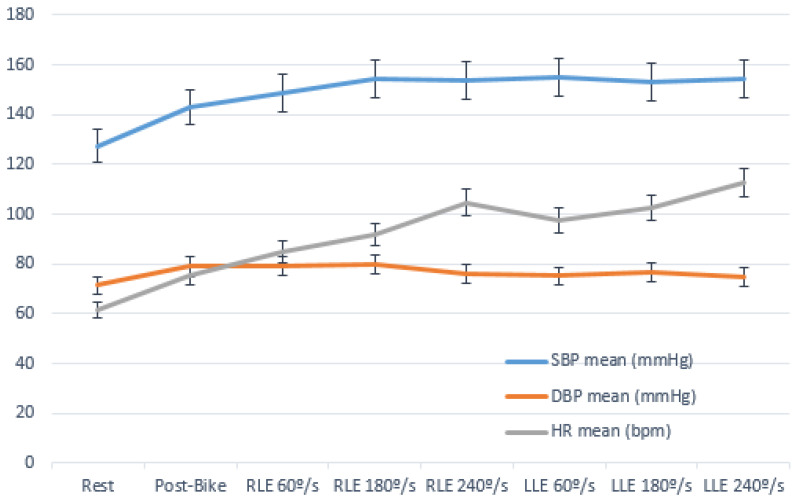
Heart rate (HR), systolic blood pressure (SBP), diastolic blood pressure (DBP) response of the global sample (n = 63) during the proposed isokinetic testing protocol (error bars show the SD of the mean in each measurement). Abbreviations: RLE, right lower extremity; LLE, left lower extremity.

**Table 1 jcm-11-01539-t001:** Description of the cardiovascular parameters during the isokinetic protocol at different measurements. Means and standard deviations obtained for the global sample (n = 63) and according to the field position and competition level of the soccer players are shown.

	Rest	Post-Bike	RLE 60°/s	RLE 180°/s	RLE 240°/s	LLE 60°/s	LLE 180°/s	LLE 240°/s
General (n = 63)								
**SBP**	127.8 ± 10.5 (152–102)	143.6± 12.4 * (181–118)	149.1 ± 14.7 *† (191–127)	155.2 ± 15.7 *† (207–119)	154.7 ± 14.3 *† (187–129)	155.2 ± 13.7 *† (189–128)	127.8 ± 10.5 (152–102)	127.8 ± 10.5 (152–102)
**DBP**	71.2 ± 8.4 (90–47)	79.3 ± 8.6 * (101–61)	79.1 ± 9.9 *‖ (103–57)	79.5 ± 10.1 ‡‖ʂ (102–58)	76.4 ± 9.8 ‡ (96–54)	75.5 ± 10.6 (96–50)	77.5 ± 10.3 ‡ (94–57)	74.9 ± 9.9 (93–44)
**MAP**	90.3 ± 7.4 (108.5–76.5)	100.7 ± 8.4 * (127.6–82)	102.4 ± 9.4 * (121.3–85.6)	104.5 ± 9.4 *† (122–86)	102.4 ± 8.8 * (125–84.6)	102 ± 9.7 * (123–78.3)	102.5 ± 10 * (124–80.3)	101.2 ± 10 * (127.6–72.3)
**HR**	61.3 ± 10.4 (92–39)	74.5 ± 13 ђ (103–45)	84.1 ± 17.5 ђ (121–48)	91.2 ± 18.4 ђ (126–49)	104.3 ± 18.7 ђ (139–65)	97 ± 19.5 ҂ (141–51)	102.3 ± 18.3 Ʉ (138–51)	112.9 ± 18.9 ђ (148–61)
**RPP**	7860.9 (11,088–4641)	10,786.1 ђ (16.109–66.72)	12,602.5 ђ (19.656–6.419)	14,189.6 ђ (21.452–7.301)	16,123.6 ђ (25.993–9.417)	15,140.5 ҂ (24.150–7.191)	15,798.1 Ʉ (23.046–7.089)	17,442.3 ђ (26.069–8.784)
**By Field Position**								
Forwards (n = 15)								
**SBP**	125.4 ± 8.4	142.6 ± 12	146.1 ± 13.3	151 ± 14.6	152.3 ± 12.4	150.1 ± 8.9	151.8 ± 13.5	153.3 ± 14.8
**DBP**	69.3 ± 8.3	76 ± 8.7	77.2 ± 8.7	74.5 ± 10	71.7 ± 8.8	72.8 ± 11	78.6 ± 9.4	74.3 ± 8.8
**MAP**	87.8 ± 7.4	98.2 ± 8	100.1 ± 8.8	99.8 ± 8	98.5 ± 8.1	98.6 ± 8.5	102.9 ± 9.7	100.6 ± 10
**HR**	61.1 ± 14	78.3 ± 17	86.9 ± 19.6	90.9 ± 17.9	106.4 ± 18.3	97.1 ± 20.2	101.2 ± 21.8	109.5 ± 23.1
**RPP**	7683.8	11,219	12,822	13,781	16,255	14,612	15,505	16,854
Midfielders (n = 25)								
**SBP**	130.8 ± 9.5	146.7 ± 12.4	153.7 ± 17	156.3 ± 15.1	156.2 ± 16.2	156.2 ± 13.7	152.9 ± 18	158 ± 16.3
**DBP**	73.3 ± 7.1	79.7 ± 8.6	79.1 ± 10.2	80.1 ± 9.9	77.4 ± 10.8	72.9 ± 10.1	74.7 ± 10.6	72.1 ± 8.4
**MAP**	92.3 ± 6.6	102 ± 9.2	103.4 ± 11.1	105.5 ± 9.6	103.7 ± 10.8	100.6 ± 10.5	100.7 ± 11.8	100.7 ± 9.9
**HR**	59.5 ± 9.2	73.6 ± 14.7	80.5 ± 15.3	87.8 ± 20	98.1 ± 20.2	90.2 ± 18.4	98.3 ± 18.3	108.4 ± 19.2
**RPP**	7815	10,854	12,458	13,779	15,425	14,151	15,075	17,148
Defenders (n = 23)								
**SBP**	126.4 ± 12.2	139.8 ± 12.5	145.2 ± 12.6	154.3 ± 13.7	152.2 ± 12.3	157.6 ± 16	154.3 ± 15.5	151.4 ± 15.8
**DBP**	71.2 ± 8.4	80.3 ± 8.6	80.4 ± 11.1	83 ± 8.8 Ω	77.9 ± 9.2	78.8 ± 10.2	77.5 ± 8.8	77.4 ± 11.2
**MAP**	89.4 ± 7.8	100.1 ± 7.8	102 ± 8.3	106.7 ± 8.6	102.6 ± 6.6	105.1 ± 9	103.1 ± 8.7	101.7 ± 10.7
**HR**	63.2 ± 8.9	74.7 ± 9.9	87.5 ± 17.8	96.5 ± 16.5	110.2 ± 16.1	105.5 ± 17.7 #	107.3 ± 14.8	119.1 ± 13.8
**RPP**	8019.6	10,436	12,731	14,855	16,705	16,605	16,503	17,925
**By Competition Level**								
1st team (n = 20)								
**SBP**	127.1 ±12.1	144 ± 14.4	149.4 ± 15.1	154.1 ± 15.3	153 ± 16.3	153.3 ± 16.3	151.6 ± 16.1	153.9 ± 15.8
**DBP**	70.8 ± 8.7	78 ± 9.9	79.4 ± 12.2	81.8 ± 8.6	75.8 ± 9.9	75.6 ± 10	77.1 ± 10	76.6 ± 9.6
**MAP**	89.4 ± 8.2	100 ± 9.6	102.7 ± 12.5	105.9 ± 7.8	101.5 ± 9.3	101.5 ± 9.6	101.9 ± 9.9	102.4 ± 10.3
**HR**	64.2 ± 12.5	76.4 ± 13.4	85.8 ± 17.2	92.1 ± 19.4	102.8 ± 20.3	96.6 ± 21	101.2 ± 16.6	103 ± 19.1
**RPP**	8127.2 ± 1598.7	10,996.7 ± 2353.8	12,801.6 ± 2753.5	14,123.5 ± 3005.2	15,723.7 ± 3608.7	14,804.7 ± 3643.6	15,329.4 ± 2978.4	15,908.8 ± 3509.4
2nd Team (n = 43)								
**SBP**	127.7 ± 9.9	142.7 ± 11.7	148.2 ± 14.9	154.3 ± 14.1	154.1 ± 12.6	156.1 ± 12.4	154.9 ± 16.5	154.1 ± 15.6
**DBP**	71.6 ± 7.6	79.5 ± 8.1	79 ± 9	78.7 ± 10.5	76.3 ± 10	74.8 ± 11.1	75.2 ± 8.5	74.4 ± 10.4
**MAP**	90.1 ± 7	100.5 ± 7.8	102 ± 9.1	103.8 ± 9.8	102.2 ± 8.7	101.9 ± 9.8	101.7 ± 9.1	100.8 ± 10.7
**HR**	59.9 ± 9.5	74.7 ± 13.7	84.2 ± 17.7	91.7 ± 18.1	105.7 ± 18.2	98.2 ± 18.9	112.7 ± 16.6	112.6 ± 20.2
**RPP**	7691.3 ± 1570.9	10,694.5 ± 2353.9	12,580.1 ± 3309	14,206.1 ± 3299.6	16,309.9 ± 3132.8	15,367.6 ± 3370.7	17,419.8 ± 3002.9	17,330.1 ± 3429.1

Data presented as mean +/− SD. Max, maximum value obtained by a sample subject; Min, minimum value obtained by a subject in the sample. Abbreviations: RLE, right lower extremity; LLE, left lower extremity; HR, heart rate; SBP, systolic blood pressure; DBP, diastolic blood pressure; MAP, mean arterial pressure; RPP, rate pressure product. * significantly higher than the resting measurement *p* < 0.001; † significantly higher than post-bike measurement *p* < 0.001; ‡ significantly higher than the measurement at rest *p* = 0.001. ‖ significantly higher than the 6th measurement (LLE 60°/s) *p* < 0.05. ʂ significantly higher than the 8th measurement (LLE 60°/s) *p* < 0.05. ђ significantly higher than all previous measurements *p* < 0.001. ҂ significantly higher than all previous measurements *p* < 0.001, except the 5th. Ʉ significantly higher than previous measurements *p* < 0.001, except for the 5th and 6th. Ω significantly higher than Forwards in the same measurement *p* < 0.05.

**Table 2 jcm-11-01539-t002:** Comparison of the cardiovascular values achieved by the global sample (n = 63) at the end of the isokinetic protocol and treadmill ergospirometry.

Variable	Rest			Final		
	Isokinetic	Ergospirometry	*p*-Value	Isokinetic	Ergospirometry	*p*-Value
SBP	127.8 ± 10.5 (152–102)	117.4 ± 8.5 (138–95)	<0.001 *	154.6 ± 14.5 (199–119)	172.4 ± 19.1 (220–120)	<0.001 *
DBP	71.2 ± 8.4 (90–47)	70.8 ± 8.2 (90–50)	0.529	74.8 ± 10.2 (93–44)	72.1 ± 12.8 (100–50)	0.279
MAP	90.3 ± 7.4 (108.5–76.5)	86.3 ± 6.8 (101.6–70)	0.814	101.2 ± 10 (127.6–72.3)	105.5 ± 11.8 (126.6–73.3)	0.044 *
HR	61.3 ± 10.4 (92–39)	60.9 ± 10.4 (92–39)	0.820	113.7 ± 19.2 (148–61)	191.5 ± 7.9 (205–173)	<0.001 *
RPP	7860.9 (11,088–4641)	7137 ± 1473 (11,960–4410)	<0.001 *	17,504 ± 3230 (26,069–8784)	33,143 ± 3852 (22,080–44,000)	<0.001 *

Data presented as mean +/− SD. HR, heart rate; SBP, systolic blood pressure; DBP, diastolic blood pressure; MAP, mean arterial pressure; RPP, rate pressure product. * Signification *p* < 0.05.

## Data Availability

The data presented in this study are available on request from the corresponding author. The data are not publicly available due to ethical considerations.

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
