# Peer review of "Blood Pressure and Heart Rate Responses to an Isokinetic Testing Protocol in Professional Soccer Players"

_jcm, 2022, doi:10.3390/jcm11061539_

Round 1

Reviewer 1 Report

Briefly summarized, this blood pressure  and heart rate  responses to an iso kinetic testing protocol in professional soccer players and compare cardiovascular parameters at completion of this isokinetic protocol with those during a treadmill test.

I hope the authors accept these comments and criticisms in the manner that was intended; that is, as an effort to offer constructive commentary and advice with a view to strengthening the manuscript:

GENERAL COMMENT 1

The design of the manuscript, data analysis and interpretation of the results are correct. However, I have several general concerns about the applicability of the findings. In what way does an isokinetic test resemble the demands of a soccer match? Why did the Authors select these protocols instead of more specific field tests? What are the real practical implications of these results? The responses of HR and BP to exercise are well known in the literature. What is the novelty of this manuscript? Please complement this justification “no reference has been found in the literature regarding cardiovascular re- 283 sponses in professional soccer players during the development of an isokinetic exercise 284 protocol for testing or training.” What is new besides the population?

According to the Authors: “The cardiovascular response of soccer players to this protocol depends on the field position but is independent of the competitive level”. However, the results do not appear strong enough to make this claim.

GENERAL COMMENT 2

The between-groups and between-categories results are not strong. As indicated by previous research, HR, Vo2max and other commonly utilized physiological parameters may not be sensitive enough to detect specific physiological adaptations occurring in response to fatigue/training (DOI: 10.1007/s00421-019-04160-3; DOI: 10.3389/fphys.2020.611550; doi.org/10.3389/fnetp.2021.711778). A feasible explanation may stem from the fact that they provide little information on the specific nonlinear dynamic interactions between organic subsystems. Therefore, it would be interesting to evaluate the effects of isokinetic/treadmill protocols utilizing variables able to quantify how respiratory, cardiovascular systems and neuromuscular systems coordinate during exercise — this analysis would probably shed some light into the differentiated physiological adaptations generated by the protocols used in this manuscript. The Authors are encouraged to briefly discuss this point in the discussion (limitations of the study or future research?).

SPECIFIC COMMENTS

  • Did the Authors perform any type of power analysis to determine the sample size?
  • Please define MAP, RPP.
  • What was the order of the tests (Isokinetic, treadmill)?
  • Line 174: The SBP, DBP, and HR values of the participants were recorded at different moments --> please indicate exactly the timing of the recordings.
  • Line 335: please correct “FC”.

Author Response

We would like to thank the reviewer for having devoted time to read, thoroughly evaluate and provide constructive criticism to our manuscript that will help to improve its overall quality.

GENERAL COMMENT 1

In what way does an isokinetic test resemble the demands of a soccer match?

Thank you for your comment. There is evidence that it is difficult to perform an accurate analysis of certain physiological values during a football match, which depends on the duration of the high-intensity phases and recovery periods, as well as the activity performed during them. Many researchers have conducted different functional assessment tests in the laboratory and controlled field tests that try to mimic as standardized and objective as possible the conditions that a football player is subjected to during a match (Chamari et al., 2004; Metaxas et al., 2005; Bangsbo et al., 2006; Aslan et al., 2012). Among all these tests, isokinetic dynamometry allows for an objective assessment of a football player's muscle function, as well as his response to maximum intensity requirements (Fousekis et al., 2011; Della Villa et al., 2012). This justification has been added in lines 56-62.

Chamari K, Hachana Y, Ahmed YB, Galy O, Sghaïer F, Chatard JC, et al. Field and laboratory testing in young elite soccer players. Br. J. Sports Med. 2004; 38: 191–6.

Bangsbo J, Mohr M, Krustrup P. Physical and metabolic demands of training and match-play in the elite football player. J. Sports Sci. 2006; 24: 665–74.

Aslan A, Acikada C, Güvenç A, Gören H, Hazir T, Ozkara A. Metabolic demands of match performance in young soccer players. J. Sports Sci. Med. 2012; 11: 170–9.

Metaxas TI, Koutlianos NA, Kouidi EJ, Deligiannis AP. Comparative study of field and laboratory tests for the evaluation of aerobic capacity in soccer players. J. Strength Cond. Res. 2005; 19: 79–84.

Fousekis K, Tsepis E, Poulmedis P, Athanasopoulos S, Vagenas G. Intrinsic risk factors of non-contact quadriceps and hamstring strains in soccer: a prospective study of 100 professional players. Br. J. Sports Med. 2011; 45: 709–14.

Della Villa S, Boldrini L, Ricci M, Danelon F, Snyder-Mackler L, Nanni G, et al. Clinical outcomes, and return-to-sports participation of 50 soccer players after anterior cruciate ligament reconstruction through a sport-specific rehabilitation protocol. Sports Health. 2012; 4: 17–24.

Why did the Authors select these protocols instead of more specific field tests?

The isokinetic test was selected as it is a test included in the functional assessment of elite football players worldwide and validated by UEFA and Spanish Association of Football Team Doctors. For this reason, this population group must face this test as a regular evaluation element of their physical condition. This explanation has been added in lines 139-142.

What are the real practical implications of these results?

Although the isokinetic assessment test is an indispensable requirement in the evaluation of the physical fitness of football players, there are no previous studies assessing the cardiac and blood pressure risk in these athletes. This statement has been added in lines 77-80.

The responses of HR and BP to exercise are well known in the literature. What is the novelty of this manuscript? Please complement this justification “no reference has been found in the literature regarding cardiovascular responses in professional soccer players during the development of an isokinetic exercise  protocol for testing or training.” What is new besides the population?

As previously mentioned, previous studies have already described the cardiac and blood pressure response of professional football players faced with other physical requirements such as isometric, concentric, or dynamic exercise. Therefore, it was considered interesting to compare the behaviour obtained in the isokinetic assessment protocol with that recorded in an ergospirometric treadmill test, whose data are widely referenced and supported in the scientific literature.

The novelty of this study, therefore, lies in describing the behaviour of BP and HR in the group of professional footballers, not because they are just another population group, but because they will be subjected to this assessment test on a regular basis during the seasons in which they compete. This means that there is a high prevalence of this test in this specific population group.  Furthermore, we understand that from the initial description of this behaviour, this test could be used in the future to obtain more global and detailed information on the blood pressure and cardiac response of a football player to these physical demands, and therefore improve the study and prevention of possible future undesirable or pathological clinical events. These ideas have been included in the Discussion section (lines 308-314).

According to the Authors: “The cardiovascular response of soccer players to this protocol depends on the field position but is independent of the competitive level”. However, the results do not appear strong enough to make this claim.

We agree with the reviewer, this sentence has been modified (lines 22-24 and 502 -505).

GENERAL COMMENT 2

The between-groups and between-categories results are not strong. As indicated by previous research, HR, Vo2max and other commonly utilized physiological parameters may not be sensitive enough to detect specific physiological adaptations occurring in response to fatigue/training (DOI: 10.1007/s00421-019-04160-3; DOI: 10.3389/fphys.2020.611550; doi.org/10.3389/fnetp.2021.711778). A feasible explanation may stem from the fact that they provide little information on the specific nonlinear dynamic interactions between organic subsystems. Therefore, it would be interesting to evaluate the effects of isokinetic/treadmill protocols utilizing variables able to quantify how respiratory, cardiovascular systems and neuromuscular systems coordinate during exercise — this analysis would probably shed some light into the differentiated physiological adaptations generated by the protocols used in this manuscript. The Authors are encouraged to briefly discuss this point in the discussion (limitations of the study or future research?).

Thank you so much for your contributions to our study. We have added this point as a limitation of our study (lines 483-493).

SPECIFIC COMMENTS

Did the Authors perform any type of power analysis to determine the sample size?

Yes, we did. Thank you for your suggestion. The analysis of minimum sample size has been added (lines 93-96).

Please define MAP, RPP.

The full term of MAP and RPP have been included in the Abstract.

What was the order of the tests (Isokinetic, treadmill)?

We have explicitly asserted this order in the line 237.

Line 174: The SBP, DBP, and HR values of the participants were recorded at different moments --> please indicate exactly the timing of the recordings.

Thank you so much for your comment. The exact timing of the recordings has been pointed out in the manuscript (lines 185-187 and 194).

Line 335: please correct “FC”.

This mistake has been corrected and replace with “HR”.

Reviewer 2 Report

Thank you for allowing me to review the following paper titled "Blood pressure and heart rate responses to an isokinetic testing 2 protocol in professional soccer players" by Perez-Gosalvez et al.

I appreciate the efforts of the authors in comparison of isokinetic testing with traditional stress testing in professional soccer players. The statistical methodology and results are well explained. I do not have any significant comments and wish all the best for the authors

Author Response

We would like to thank the reviewer for having devoted time to read, thoroughly evaluate and provide constructive criticism to our manuscript that will help to improve its overall quality.

Round 2

Reviewer 1 Report

The manuscript has been significantly improved.